# Antioxidant Activity, Formulation, Optimization and Characterization of an Oil-in-Water Nanoemulsion Loaded with Lingonberry (*Vaccinium vitis-idaea* L.) Leaves Polyphenol Extract

**DOI:** 10.3390/foods12234256

**Published:** 2023-11-24

**Authors:** Siyu Wang, Yuan Cheng, Jingyi Wang, Miao Ding, Ziluan Fan

**Affiliations:** 1School of Forestry, Northeast Forestry University, 26 HeXing Road, XiangFang District, Harbin 150040, China; siyuwang1122@163.com (S.W.); cy_lyz1109@163.com (Y.C.); owo_guyu@163.com (J.W.); 18724457575@163.com (M.D.); 2Key Laboratory of Forest Food Resources Utilization, Harbin 150040, China

**Keywords:** nanoemulsion, response surface methodology, antioxidant activity, lingonberry (*Vaccinium vitis-idaea* L.), delivery system

## Abstract

The active ingredients in lingonberry leaves and their beneficial properties to the human body have been well confirmed. In order to improve the stability and antioxidant activity of the active ingredients in lingonberry leaves, the response surface optimization method was used to prepare an oil-in-water nanoemulsion of polyphenol extract from lingonberry leaves. The active components in the extract were analyzed by ultra-performance liquid chromatography with triple quadrupole mass spectrometry (UPLC-TQ-MS), and bioactive compounds such as apigenin, sorbitol, and hesperidin were mainly found. Nanoemulsion droplets of 120 nm in diameter were prepared using ultrasonic emulsification. The optimal nanoemulsion formulation was determined through rigorous testing, and it was determined to be 10% (*w*/*w*) lingonberry extract and 20% (*w*/*w*) medium chain triglyceride (MCT). Additionally, a surfactant mixture was used, which combined soy protein isolate (SPI) and whey protein isolate (WPI) at 4% (*w*/*w*). The preparation method utilized ultrasonic emulsification, applying an ultrasonic power of 360 W for a duration of 300 s. The antioxidant activity (DPPH inhibition rate, ABTS inhibition rate and total reducing power) of the lingonberry nanoemulsion was significantly higher than that of the lingonberry polyphenol (LBP) extract. The nanoemulsion prepared using the optimal formulation had an entrapping efficiency of 73.25% ± 0.73% and a diameter of 114.52 ± 0.015 nm, with a satisfactory particle size of nanoscale and a PDI of 0.119 ± 0.065, demonstrating good stability of the emulsion.

## 1. Introduction

Lingonberry (*Vaccinium vitis-idaea* L., family *Ericaceae*) is a natural wild berry with significant nutritional value and a wide range of health-promoting properties; lingonberry is rich in phenolics, which are the main active substances. Lingonberry, often termed a “superfruit” is renowned for its substantial antioxidant properties. Predominantly found in regions of northern and central Europe, Canada, and Asia, including mountainous zones like Northeastern China’s Greater Khingan and the Lesser Hinggan Ranges, it is a source of various potent bioactives [1]. While bilberry and lingonberry leaves emerge as primary by-products during berry collection, recent studies have highlighted an intriguing observation [2], i.e., the leaves and stems from the Vaccinium species house a more substantial phenolic content than the berries themselves. The elevated phenolic concentration is congruent with their superior antioxidant capabilities compared to the fruit [3]. Such findings suggest that these by-products could be harnessed as promising reservoirs of bioactive compounds, opening doors for their inclusion in food supplements, nutraceuticals, or functional food products.

Regarding the biochemical makeup of lingonberry leaves, they feature a plethora of compounds, including phenolic acids, anthocyanins, triterpene acids, organic acids, carbohydrates, glycosyl hydroquinones, and proanthocyanidins [4]. This rich profile renders the leaves a more chemically diverse entity than the berry fruits. Notably, recent research has spotlighted flavonoid glycosides as the dominant phenolic entities in lingonberries. These compounds exhibit a myriad of health-promoting attributes, such as astringent, antitussive, diuretic, neuroprotective, antioxidant, and anti-inflammatory activities. Their potential roles in inhibiting cancer cell proliferation further underscore their therapeutic potential [5].

In addition these properties, there have been studies that have indicated that extracts from lingonberry leaves can curtail hepatitis C virus expression and stymie the proliferation of sensitive human promyelocytic leukaemia HL60 cells. Historically and traditionally, lingonberry leaves have been employed to mitigate cholesterol, address gastric complications, treat rheumatic ailments, and combat infections in the bladder and kidneys [6,7,8]. The various plant parts of the lingonberry are used in the production of pharmaceuticals, cosmeceuticals, and nutritional foods. The fruit is widely consumed and used in the food industry, while the leaves are an excellent source of phenolic compounds, especially functional products containing proanthocyanidins.

However, leaves have always been the raw materials that are not fully developed and utilized by people. Studies have shown that the leaves of lingonberry have higher biological activity than the fruit, making the leaves a promising medicinal resource [9]. They are an ignored renewable resource with huge reserves; the active ingredients in lingonberry leaves have broad application prospects.

Lingonberry polyphenols have antioxidant and free-radical scavenging properties, and their antitumour and anticancer effects should not be underestimated. However, most polyphenols have poor solubility, low stability, easy metabolism and excretion, and extremely low bioavailability, making it difficult for them to exert their effects and limiting their applications in functional foods and drugs.

Nanoemulsions have the characteristics of small particles, large specific surface area, and relatively uniform distribution, which can protect biologically active and unstable substances by encapsulating them in emulsions, and therefore the active substances can maintain their original biological activity for a long time. The small particle size, kinetic stability, and high optical transparency of nanoemulsions compared to conventional emulsions, as well as their ability to adjust the texture of a product, give them an advantage in many technical applications. Currently, nanoemulsions loaded with polyphenols have a bright future with industrial production possibilities in the field of pharmaceuticals and functional foods. Nanoemulsions are generally valued by researchers as a new type of carrier system and are increasingly being used in the fields of biology, medicine and pharmacy, and functional food development. As new active substance carriers, nanoemulsions are mainly formed by dispersion and homogenization followed by mixing water, oil, and emulsifier in appropriate proportions to form thermodynamically stable homogeneous dispersion systems. Nanoemulsions are often set to have specific functional properties, e.g., controlled release, enhanced bioavailability, enhanced potency, synergistic or targeted release, thermodynamic stability, and easy preservation; due to their small particle size and low surface tension, the active substance is more likely to come into direct contact with the gastrointestinal epithelium and promote absorption. Several studies have successfully constructed the delivery system of polyphenol nanoemulsions, improving stability [10,11], biological accessibility [12,13,14], antioxidant [15,16], and other functional activities of polyphenols. Nanoemulsions have been widely used to improve the bioavailability of insoluble drugs.

It is imperative to note that nanoemulsions are not equilibrium systems. They are non-thermodynamically stable and do not form spontaneously. Their formation is influenced either by their inherent chemical potential within the emulsion system or by external forces. Emulsification methods, which are pivotal to their formation, can be broadly classified based on the energy source utilized. There are high-energy methods that rely on external energy and low-energy methods that harness the system’s inherent potential.

Among the high-energy methods, techniques like ultrasound, micronization, and high-pressure homogenization are prevalent. Notably, ultrasound technology stands out due to its efficiency and cost-effectiveness and can deliver nanoemulsions with smaller droplet size, reduced polydispersity index (PDI), and superior stability, all while using fewer surfactants. Ultrasound technology employs a frequency of at least 20 kHz [17]. The ultrasonic vibrations produce pronounced shear and pressure gradients in the emulsion. This facilitates the breakdown of larger droplets, enabling continuous production of the nanoemulsion. Notably, the energy consumption and associated costs are considerably lower than those in high-pressure homogenization.

To conclude, as the quest for optimal drug delivery systems persists, nanoemulsions are progressively recognized for their multifaceted benefits, particularly in the pharmaceutical and functional food industries. Their potential in enhancing the delivery and efficacy of compounds like LBPs is undeniable [18]. The total experimental content of this study is shown in Figure 1.

## 2. Materials and Methods

### 2.1. Materials

SPI, WPI, and MCT were purchased from Sigma (Shanghai, China). All other chemicals and reagents used were of analytical grade.

The plant material was acquired from Kudu Forestry Bureau, Hulunbeir, Inner Mongolia Autonomous Region. Picking was conducted during the harvest of lingonberry plants, in the middle of September. Then, the plant material was transported to the laboratory, where it was cleaned of residual soil, and the different parts were separated, placed in a container, and stored them in a freezer. Subsequently, the samples were lyophilized using a freeze dryer (SP Scientific, Gardiner, MT, USA).

### 2.2. Preparation of Lingonberry Leaves Extract

Taking into consideration edible safety and the foundation established by our research team [19], we employed food-grade ethanol for the solvent extraction of polyphenols. The extraction procedure was adapted from the methodology presented in [20], with minor modifications. First, the freeze-dried lingonberry leaves were pulverized into a fine powder using an FW-100 high-speed universal grinder (Braun, Frankfurt, Germany), and then sieved using 100 mesh filter, freeze-stored at −20 °C in a refrigerator, and sealed for storage.

Then, 400 mL of 75% precooled ethanol was added and mixed with an RZ-8012 handheld stirrer at 12,000 r/m for 5 min. The prepared extract was homogenized (BME 100 L, Weiyu, Shanghai, China) with a high shear for 3 min and stored at −20 °C in a refrigerator prior to preparation of the nanoemulsion.

### 2.3. Characterization of LBPs by UPLC-TQ-MS

The extract of lingonberry leaves, prepared as described in Section 2.2, was evaporated to dryness using a rotary evaporator at 45 °C under vacuum conditions. Then, the resulting extract was redissolved in 5 mL of methanol and subsequently filtered through a 0.45 μm filter prior to analysis. Identification of phenolic compounds in the extract was performed on an UPLC-TQ-MS system (Waters ACQUITY UPLC^TM^ system, Waters, Milford, CT, USA). The separation of phenolic compounds was conducted in a chromatographic column, Hyperil Glod (100 × 2.1 mm), with a temperature of 25 °C. Ionization Polarity: ESI− and ESI+; spray voltage, 3800/3000 V (+/−); capillary temperature, 320 °C. The positive mobile phase (solvent A) was prepared using 0.1% formic acid (FA) in water and MeOH, while the negative phase (solvent B) was created using 5 mM ammonium acetate in water (ammonia water regulation, pH = 9) and MeOH. The flow rate was set to 0.35 mL/min. Data obtained from UPLC-TQ-MS were analyzed using the Compound Discoverer 3.2 software.

### 2.4. Preparation of the Nanoemulsion

An ultrasonic homogenizer (ASU-10D, AS one, Matsubara, Japan) was used to produce the nanoemulsion samples. Two natural emulsifiers, soybean isolate and whey isolate, were used as the emulsifiers. A single-factor test was conducted with the mass concentration of emulsifier, oil phase type, oil phase mass fraction, protein ratio, and extract mass fraction as the observed factors. The encapsulation rates of lingonberry nanoemulsion samples prepared under different conditions were determined separately.

Based on the results of a single-factor test, the emulsifier mass concentration, oil phase mass fraction, and extract mass fraction were selected as the three factors to design a response surface test to optimize the formulation of the lingonberry nanoemulsion. Based on the response surface Box–Behnken design principle, the Design-Expert 8.0 software was used for the design and analysis of the response surface test. A three-factor, three-level response surface analysis was used to design the test using the encapsulation rate of lingonberry nanoemulsion as the response value, and the factor levels are shown in Table 1.

The RSM with BBD experimental design with 3 levels, 3 independent variables, and 17 runs including five center points (Table 2) was used to optimize the operation condition including.

Mass concentration of emulsifier (X1), oil phase mass fraction (X2), and extract mass fraction (X3) of LBP nanoemulsions (LBPNs) were determined using the Design-Expert software (Version 8.0.6, Stat-Ease Inc., Minneapolis, MN, USA):(1)$$y=\beta_{0}+\sum_{i=1}^{3}\beta_{i}X_{i}+\sum_{i=1}^{3}\beta_{ii}X_{i}^{2}+\sum_{i=1}^{2}\sum_{j=i+1}^{3}\beta_{ij}X_{i}X_{j}+\varepsilon$$

In the present model (Equation (1)), *X*1, *X*2, and *X*3 are defined as independent variables, whereas *Y* represents the dependent response variable, which is assessed for each unique combination of factorial levels. The model comprises various coefficients: *β*_0_ (intercept), *β_i_* (linear), *β_ii_* (quadratic), and *β_ij_* (interaction coefficients). These coefficients of the second-order polynomial model are estimated through a multiple regression analysis based on experimental response values. Model fit was evaluated using several metrics as follows: the coefficient of determination (R^2^), adjusted coefficient of determination (R^2^_adj_), *p*-value, and the degree of fit. The model was considered to be a well-fitting model when R^2^ approaches 1 and the lack-of-fit *p*-value exceeds 0.05, signifying statistical insignificance. Additionally, the F-test is employed to ascertain the statistical significance of R^2^, while regression coefficients’ statistical relevance is determined using the *t*-test [21].

### 2.5. Characterization of the W/O Nanoemulsion

#### 2.5.1. Encapsulation Efficiency

The encapsulation efficiency was performed using the Folin–Ciocalteu method described by Mahsa Yazdan-Bakhsh et al. Free surface phenolic compounds were extracted by adding 5 mL of n-hexane to 1 mL of nanoemulsion and centrifuged for 3 min using a centrifuge at 3500× *g*. Then, 5 mL of ethanol was added to 1 mL of nanoemulsion and the total phenolic compounds were released using ultrasonic shaking for 1 h to disrupt the emulsion encapsulation structure. The encapsulation efficiency was calculated as the difference between the total phenolic content in the nanoparticle suspension and the TPC content in the supernatant [22].

#### 2.5.2. Particle Size and PDI

The mean droplet diameter and polydisperse index (PDI) were measured utilizing a Nano-ZS PALS laser particle size analyzer (Malvern Instruments Ltd., Malvern, UK). The measurements were conducted through the dynamic light scattering method, operational at a light wavelength of 660 nm and a scattering angle fixed at 90°, under a controlled temperature of 25 °C ± 0.1 °C. The PDI, a non-dimensional metric, varies from 0 to 1, with a refractive index set at 1.330 for these measurements.

#### 2.5.3. Zeta Potential

The zeta potential (ZP) values of the samples were ascertained using a Nano-ZS PALS laser particle size analyzer (Malvern Instruments Ltd., Malvern, UK). Prior to analysis, the samples were diluted to a standard concentration of 5 mg/mL with water. This dilution was necessary to minimize the effects of multiple scattering during measurements.

#### 2.5.4. Confocal Laser Scanning Microscopy (CLSM)

The LBPN aqueous and oil phase embedding states were evaluated using a confocal laser scanning microscope (Leica, Heidelberg, Germany) with 40× magnification at 20 °C, and the proteins were treated with Nile Blue (1 mg/mL in anhydrous ethanol, 1:100 *v*/*v*) fluorescent dye solution, followed by Nile Red (1 mg/mL in dimethyl sulfoxide, 1:100 *v*/*v*). The oil stain treatment was performed according to the method described by [23]: 100 μL of emulsion was mixed with 10 μL of Nile Red and 10 μL of Nile Blue and left for 10 min, and then the proteins were detected using Ar/K and He/Ne dual-channel laser modes with an excitation light of 633 nm, and then the excitation light was changed to 488 nm to detect the oil phase. The software used for the CLSM imaging was ImageJ 2.0.

### 2.6. Antioxidant Activity of LBPs

#### 2.6.1. DPPH Radical Scavenging Assay

For the DPPH (0.2 mmol/L) assays, the solution was prepared in absolute methanol and stored at 4 °C in darkness. To execute the test, 0.1 mL of ascorbic acid or a sample variant was mixed with 1.5 mL of the DPPH solution. Then, the mixture was incubated in darkness for 30 min. Ascorbic acid’s absorbance was monitored at 515 nm, serving as the control (*A_control_*). The DPPH radical scavenging activity of the sample was calculated as follows:DPPH radical scavenging activity(%) = (1 − *A_sample_*/*A_control_*) × 100(2)

#### 2.6.2. ABTS•+ Radical Scavenging Assay

The ABTS solution was formulated by combining an ABTS stock solution (7 mmol/L) with K_2_O_8_S_2_ (142 mmol/L) at a 1:1 ratio. Then, the solution was stored in a light-deprived environment at ambient temperature for 12 h. Subsequently, 100 μL of the sample was introduced to 2.9 mL of the ABTS+ solution. After mixing, the combined solution was shielded from light and left to react for 6 min at room temperature. The resulting absorbance was measured at 734 nm using a spectrophotometer, with ascorbic acid serving as the control reference (A_control_). The ABTS radical scavenging activity of the sample was calculated as follows:ABTS radical scavenging activity(%) = (1 − A_sample_/A_control_) × 100(3)

#### 2.6.3. Reducing Power

The reducing power of the LBPs and LBPN were assessed following the protocol outlined by [24]. A volume of 2.5 mL from the sample, at differing concentrations, was amalgamated with 2.5 mL of potassium hexacyanoferrate buffer (0.2 M, pH 6.6) and 2.5 mL of potassium ferricyanide (1%). The mixture was subjected to an incubation period at 50 °C for 20 min. Post incubation, 2.5 mL of 10% trichloroacetic acid, 2.5 mL of distilled water, and 0.5 mL of 0.1% ferric chloride solution were introduced. The mixture’s absorbance was gauged at 700 nm employing a spectrophotometer.

### 2.7. Statistical Analysis

Each analysis was executed three times, with the resultant data articulated as means ± standard deviation (SD). Notable differences across data sets were scrutinized through the analysis of variance (ANOVA) facilitated by the SPSS Statistics software, version 20 (IBM, New York, NY, USA).

## 3. Results

### 3.1. UPLC-TQ-MS Characterization of Extract

The polyphenol extraction rate was calculated based on the method outlined in [25] and was found to be 138.18 mg/g. In this study, in the leaves of lingonberry, 14 phenolic compounds were identified, originating from three phenolic groups: phenolic acids, flavonoids, and terpenoids. The identified 14 compounds are listed in Table 3. The three most abundant active substances in the lingonberry leaves were apigenin 7-rhamnosyl-(1->2)-galacturonide, Bis(4-ethylbenzylidene)sorbitol, and pinocembrin7-O-neohesperidoside3-O-acetate, all belonging to the class of flavonoids. The second most abundant substance in the leaves was a sort of phenolic acid, i.e., caffeic acid. In this study, no anthocyanin substances were identified, possibly due to the anthocyanin substances have been lost due to the long freezing time of the raw material. Nevertheless, in previous reports, a variety of anthocyanins such as cyanidin-3-O-arabinoside, cyanidin-3-O-glucoside, and delphinidie-3-glucoside have been found in lingonberry leaves [26]. According to previous reports, the presence of anthocyanins in lingonberry leaves may be related to the variety, origin, freshness, phenological stage, and ripeness of lingonberries [27]. Anthocyanins were detected only in lingonberry fruits and not in leaves when phenolic substances of ten different varieties of lingonberry such as ”Erntekrone”, ”Koralle”, and ”Masovia” were analyzed [28]. At the same time, one study reported that anthocyanins were not detected in three types of lingonberry leaves collected from Romania [26]. In contrast, active substances in roots, stems, leaves, and fruits of lingonberries harvested from Inner Mongolia, China were analyzed and two anthocyanins were detected, i.e., empetrin and 3,5-diglucosyldelphinidin [29].

Apigenin 7-rhamnosyl-(1->2)-galacturonide, which is most abundant in the leaf extract, is an apigenin derivative. Apigenin is a secondary plant metabolite, usually found in nature in glycosylated form, and is one of the most abundant and well-studied flavonoids. It can be used as a cancer chemopreventive agent. It has been shown that apigenin-7-O-glucoside is more effective than apigenin in reducing colon cancer cell viability and inducing cell death, and that apigenin-7-O-glucoside is more biologically active than apigenin [30]. With the increasing attention paid to plant-derived diets, the application of natural active substances such as apigenin in food has research value.

The high content of the soluble sugar sorbitol in the extract may be due to the fact that sorbitol is the main form of carbohydrate accumulation in lingonberries. Previous studies have shown that sorbitol is the major sugar component of water-extracted chokeberry leaves, accounting for ca. 80% of the total carbohydrates, with an average content of 145.2 ± 1.8 mg/g in the lyophilized leaves [31].

The higher content of caffeic acid also provides antioxidant activity to the extract. Caffeic acid, as a natural plant-derived antioxidant, has gradually become a hot research topic in recent years [32] with studies conducting in vivo and in vitro antioxidant tests. In in vitro experiments, caffeic acid has shown stronger antioxidant activity than chlorogenic acid, and it has been reported that caffeic acid may play a major role in the protective effect of chlorogenic acid against ischemia reperfusion injury. In addition to the aforementioned active substances, the extract also contains a relatively abundant variety of phenolic compounds, including quercetin, oleanolic acid, and various other phenolic compounds, totaling 14 different types. These findings are in line with previous research on the polyphenolic content of lingonberry leaves [20,33].The total ion current of LBPs is illustrated in Figure 2.

### 3.2. Optimization of the Lingonberry Nanoemulsion Using Response Surface Methodology

A response surface analytical approach was utilized, taking into consideration protein concentration, oil phase mass fraction*,* and extract mass fraction as independent variables, while the entrapping efficiency was designated as the dependent variable. From this analysis, a regression equation was derived as:Entrapping efficiency = +73.12 + 0.51 × *A* − 1.28 × *B* − 3.05 * *C* + 0.65 × *A* × *B* − 0.51 × *A* × *C* + 4.01 × *B* × *C* − 8.45×*A*^2^ − 9.37 × *B*^2^ − 11.09 × *C*^2^(4)
where A is the protein concentration, B is the oil phase mass fraction, and C is the extract mass fraction.

Table 4 demonstrates that the model’s *p*-value is below 0.0001, signifying statistical significance. Conversely, the lack-of-fit item has a *p*-value of 0.9364, denoting an insignificant level, which suggests the validity of the response surface model. The quadratic model obtained was fitted with the data for the responses. Figure 3 illustrates the normal probability plot residuals for the entrapping efficiency. The majority of the results align closely with a straight trajectory, insinuating a normal distribution of the outcomes and their compatibility with the regression model. Consequently, this response surface model adeptly captures the correlation between diverse parameters and the response metric, rendering it suitable for guiding the LBPN preparation phase process. Figure 4, wherein two factors are held constant to assess the interplay of the remaining factors, evidences that maintaining one factor static causes variations in the other two factors, affecting the entrapping efficiency. This negates the possibility of a mere linear correlation between factors and entrapping efficiency. As a result, the response surface methodology was employed to fine tune and discern the paramount conditions for LBPN synthesis. This optimization revealed that at a protein concentration of 4.03%, oil phase mass fraction of 19.50%, and extract mass fraction of 9.22%, the zenith of entrapping efficiency was achieved at 73.4329%. For ensuing studies, given the experiment’s feasibility, the conditions were slightly adjusted, i.e., protein concentration to 4%, oil phase mass fraction to 20%, and extract mass fraction to 10%. The entrapping efficiency of the revised LBPN closely mirrored the predictions from the response surface model, cementing the model’s precision and dependability. Previous studies have utilized maltodextrin and glucose microcapsules as carriers to prepare lingonberry microcapsules, achieving microencapsulation rates of 79–81% [34].

### 3.3. Characterization of the Optimal Lingonberry Nanoemulsion

#### 3.3.1. Particle Size and PDI

The LBPN was prepared with the formulation optimized using the response surface methodology, and the entrapping efficiency of the LBPN was close to 73%, with 114.52 ± 0.015 nm and 0.119 ± 0.065 for average particle size and PDI, respectively. As depicted in Figure 5, the average droplet dimension of the LBPN formulation was observed to be within the nanometric domain (<500 nm). In terms of the PDI, a PDI metric below 0.2 signifies that the particles within the nanoemulsion exhibit monodispersity [35].

#### 3.3.2. Zeta Potential

Figure 6 reveals that the optimally developed LBPN demonstrated a zeta potential of −23.29 ± 0.4471 mV. The negative zeta potential values for the optimized samples suggest that they are negatively charged, contributing to their physical stability. Zeta potential is integral to colloidal dispersion stability as it reflects the magnitude of repulsion among adjacent particles. Enhanced electrostatic repulsion among the droplets in this context prevents the coalescence of the emulsion particles, thereby stabilizing the system [36].

#### 3.3.3. Confocal Laser Scanning Microscopy (CLSM)

The microstructure of LBPN is depicted in Figure 7 as shown. The protein and oil phases were labeled with Nile blue and Nile red, respectively, and the SPI/WPI mixed protein phase was green in the image, while the MCT was red. From CLSM imaging, it can be seen that the LBPN emulsion particles are in the shape of regular spheres, with each droplet dispersed in rows, and the mixed proteins wrapped around the oil droplets to form an oil-in-water emulsion structure. It has been shown that fresh emulsions prepared using ultrasonic emulsification have very fine and uniformly dispersed oil droplets, demonstrating that ultrasound is an effective way to prepare emulsions. The high overlap between the green protein phase and the red oil phase proved that the hybrid protein phase of the WPI/SPI was effectively combined with the MCT, indicating good emulsifying ability of the hybrid emulsifier WPI/SPI [37].

#### 3.3.4. Antioxidant Activity

The graphic (Figure 8) representation clearly indicates that nanoemulsification substantially augments the DPPH antioxidant activity of LBPs. The ability of the LBPN to quench the DPPH radical was evident through the transition from purple to yellow, a change that was further validated by the observed decrease in absorbance at 515 nm. As depicted in Figure 8, there was a pronounced decrease in the DPPH radical, attributed to the scavenging capacity of the resveratrol solution, the nanoemulsion, and the benchmark antioxidant, ascorbic acid. The percentage inhibition obtained was maximum for ascorbic acid (85.43% ± 2.85%) compared to the LBPN (79.37% ± 3.18%) and LBPs (63.58% ± 2.95%). The utilization of nanoscale emulsions for encapsulating polyphenols greatly enhances their solubility, thus optimizing the antioxidative properties of polyphenols. Research has shown that the preparation of resveratrol as a resveratrol nanoscale emulsion resulted in a DPPH inhibition rate as high as 83.93 ± 3.81%, an increase of nearly 13% compared to non-encapsulated resveratrol solutions [15]. Kumar et al. [11] employed lecithin and Tween-80 as emulsifiers to prepare a resveratrol-loaded nanoscale emulsion, which remained stable without phase separation even after four months of storage. Under ultraviolet radiation exposure, the nanoscale emulsion significantly slowed down and inhibited the degradation of resveratrol.

Regarding the ABTS assay, the antioxidant activity of the LBPN was markedly superior to that of LBPs. The ABTS assay assesses antioxidant capacity via a single electron transfer mechanism, gauging the reduction of the ABTS radical cation. The incorporation of whey protein isolate (WPI) as a natural emulsifier in the formulation likely enhanced the LBPN’s antioxidant properties. The presence of hydrophobic amino acids like alanine, isoleucine, valine, and those with aromatic side chains in the WPI plays a pivotal role in the radical scavenging activities of the protein. Additionally, the homogenization process may have facilitated a more effective dispersion of LBPs within the nanoemulsion, potentially elevating its antioxidant efficacy [38].

The reduction capability of both the drug solution and the resveratrol nanoemulsion was benchmarked against ascorbic acid (standard). The reductive ability of the LBPN was evaluated based on the direct reduction of Fe[(CN)_6_]_3_ to Fe[(CN)_6_]_2_. Such reducing properties of a compound typically indicate its antioxidant potential, suggesting that the LBPN may serve as an effective antioxidant. Notably, as depicted in Figure 8, the reducing strength of the LBPN surpassed that of LBPs.

## 4. Conclusions and Future Research

In conclusion, this study successfully developed a lingonberry polyphenol extract-loaded oil-in-water nanoemulsion using a response surface optimization method. The nanoemulsion exhibited small droplet size, good stability, and enhanced antioxidant activity compared to the extract. These findings contribute to the understanding of the formulation and optimization of nanoemulsions for the delivery of bioactive compounds.

Moving forward, further research can explore the potential applications of this lingonberry nanoemulsion in the food and pharmaceutical industries. The optimized formulation can be utilized to enhance the stability and bioavailability of bioactive compounds, leading to improved functional food products and nutraceuticals. Additionally, this study can be extended to investigate the in vivo efficacy and safety of the nanoemulsion, providing valuable insights for future clinical applications.

Furthermore, it is important to consider the practical implications of this research. The development of a stable and bioactive nanoemulsion opens up opportunities for the utilization of lingonberry polyphenols in various products, such as functional beverages, dietary supplements, and topical formulations. However, it is crucial to address the potential limitations and challenges associated with the scale-up and commercialization of the nanoemulsion, including cost-effectiveness, regulatory considerations, and consumer acceptance. Nanocarriers have been primarily used in drug delivery systems and the development of functional foods. It is, therefore, essential to delve deeper into studies concerning their toxicity and underlying mechanisms of any toxic effects, in order to determine the permissible daily intake of nanoemulsions.The absorption and metabolic processes of nanoscale carriers differ from those of microscale carriers. Currently, the potential toxicity of nanoscale emulsions upon absorption by the human body remains unclear and requires further in-depth investigation, such as a systematic evaluation of the potential risks of the LBPN through toxicological experiments, including clinical studies.

In summary, this study lays the foundation for the application of lingonberry polyphenols in nanoemulsion formulations, highlighting their potential for improving stability, bioavailability, and antioxidant activity. Further research and development efforts are warranted to fully explore the practical applications and benefits of this technology in the food and pharmaceutical industries.

## Figures and Tables

**Figure 1 foods-12-04256-f001:**
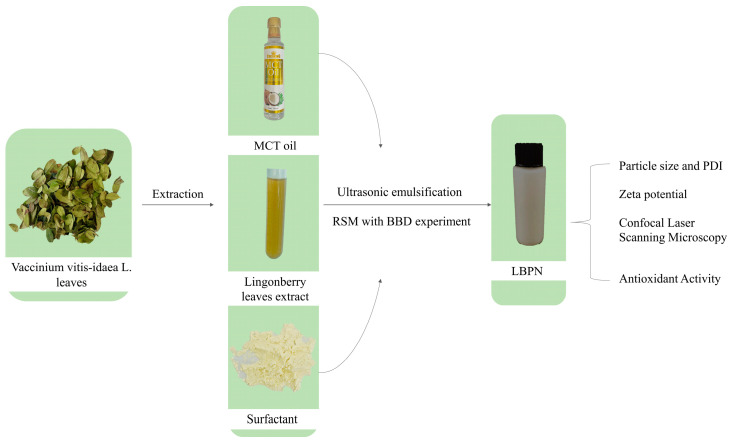
The total experimental contents.

**Figure 2 foods-12-04256-f002:**
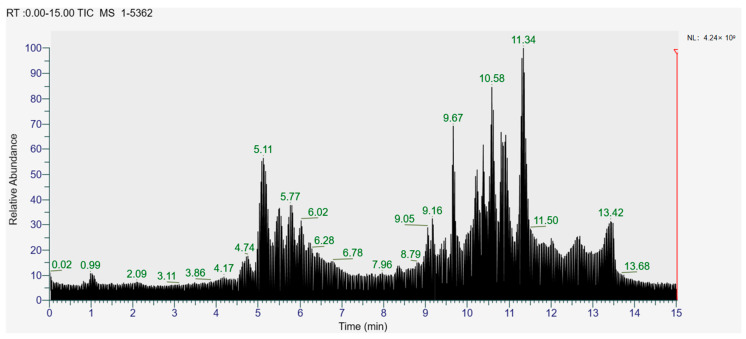
Total ion current (TIC) of LBPs.

**Figure 3 foods-12-04256-f003:**
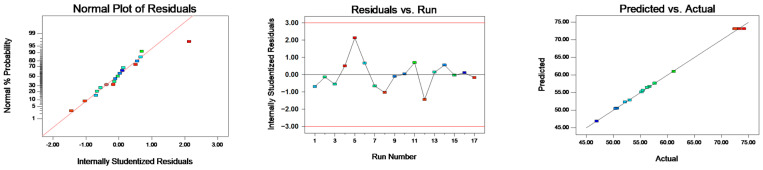
Normal probability plot residual for entrapping efficiency.

**Figure 4 foods-12-04256-f004:**
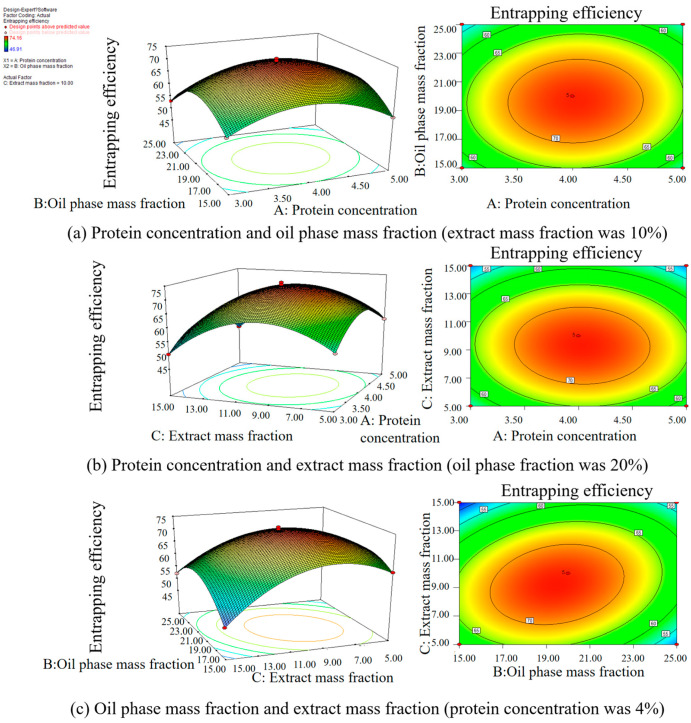
Effect of two factors on the entrapping efficiency of the LBPN.

**Figure 5 foods-12-04256-f005:**
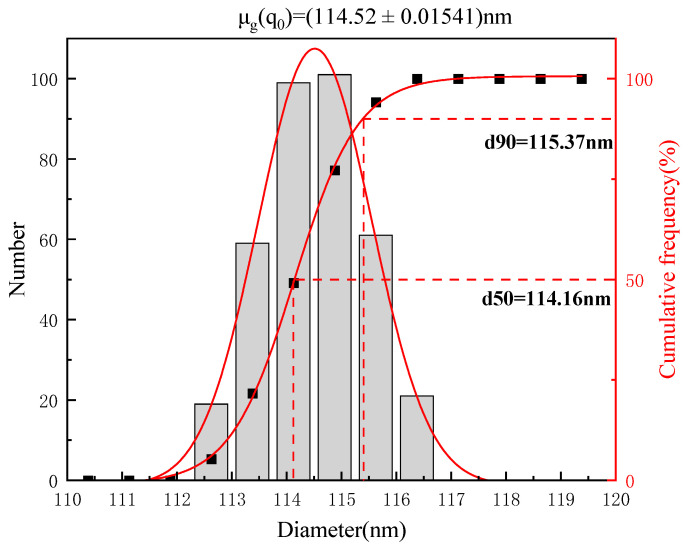
Particle size distribution of the optimally formulated LBPN.

**Figure 6 foods-12-04256-f006:**
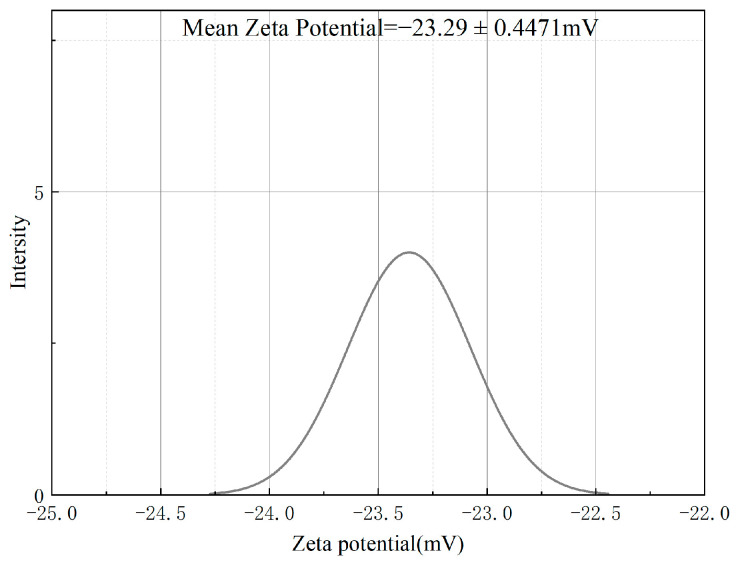
ζ-potential of the optimally formulated LBPN.

**Figure 7 foods-12-04256-f007:**
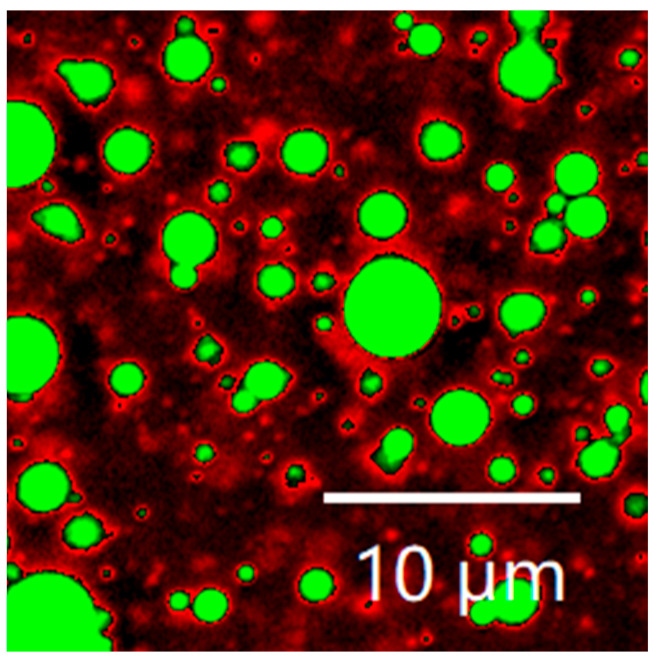
Confocal laser scanning microscopy image of the optimally formulated LBPN.

**Figure 8 foods-12-04256-f008:**
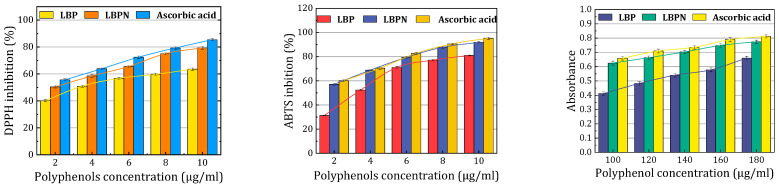
Comparison of antioxidant properties of the LBPN, LBPs, and ascorbic acid.

**Table 1 foods-12-04256-t001:** Box–Behnken experimental factors and levels.

Independent Variable		Actual Levels at Coded Factor Levels	
	−1	0	1
Mass concentration of emulsifier	3%	4%	5%
Oil phase mass fraction	15%	20%	25%
Extract mass fraction	5%	10%	15%

**Table 2 foods-12-04256-t002:** Box–Behnken design.

Run	Protein Concentration (X1,%)	Oil Phase Mass Fraction (X2,%)	Extract Mass Fraction (X3,%)
1	4.00 (0)	25.00 (1)	15.00 (1)
2	3.00 (−1)	15.00 (−1)	10.00 (0)
3	3.00 (−1)	20.00 (0)	5.00 (−1)
4	4.00 (0)	20.00 (0)	10.00 (0)
5	4.00 (0)	20.00 (0)	10.00 (0)
6	3.00 (−1)	25.00 (1)	10.00 (0)
7	5.00 (1)	15.00 (−1)	10.00 (0)
8	4.00 (0)	20.00 (0)	10.00 (0)
9	4.00 (0)	25.00 (1)	5.00 (−1)
10	3.00 (−1)	20.00 (0)	15.00 (1)
11	4.00 (0)	15.00 (−1)	5.00 (−1)
12	4.00 (0)	20.00 (0)	10.00 (0)
13	5.00 (1)	25.00 (1)	10.00 (0)
14	5.00 (1)	20.00 (0)	15.00 (1)
15	5.00 (1)	20.00 (0)	5.00 (−1)
16	4.00 (0)	15.00 (−1)	15.00 (1)
17	4.00 (0)	20.00 (0)	10.00 (0)

**Table 3 foods-12-04256-t003:** The main phenolic compound contents in the leaves of lingonberry using UPLC-TQ-MS.

Chemical Name	Area	RT/min	Molecular Weight	Molecular Formula	Molecular Structure Formula
Apigenin 7-rhamnosyl-(1->2)-galacturonide	1.73 × 10^9^	12.033	592.14407	C_27_H_28_O_15_	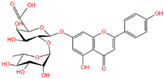
Bis(4-ethylbenzylidene)sorbitol	1.26 × 10^9^	9.157	414.19921	C_24_H_30_O_6_	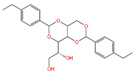
Pinocembrin 7-O-neohesperidoside 3-O-acetate	6.48 × 10^8^	12.596	606.19569	C_29_H_34_O_14_	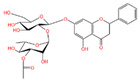
Caffeic acid	4.49 × 10^8^	9.662	162.02992	C_9_H_8_O_4_	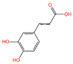
Licochalcone A	2.27 × 10^8^	6.201	360.13567	C_21_H_22_O_4_	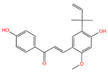
Munduserone	1.76 × 10^8^	0.987	342.11153	C_19_H_18_O_6_	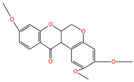
Kaempferol-3-O-α-L-arabidopyranoside	1.29 × 10^8^	7.316	418.0876	C_20_H_18_O_10_	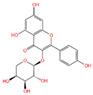
Quercetin-3β-D-glucoside	1.05 × 10^8^	7.135	464.08967	C_21_H_20_O_12_	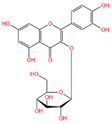
Oleanolic acid	9.03 × 10^7^	10.561	456.35435	C_30_H_48_O_3_	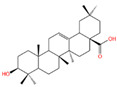
Cycloartomunoxanthone	5.21 × 10^7^	5.695	448.15239	C_26_H_24_O_7_	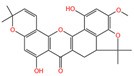
Quercetol B	1.60 × 10^7^	10.416	368.19811	C_23_H_28_O_4_	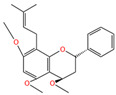
Cycloartocarpin	1.58 × 10^7^	6.83	434.17079	C_26_H_26_O_6_	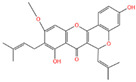
Silybin	1.39 × 10^7^	10.676	482.12114	C_25_H_22_O_10_	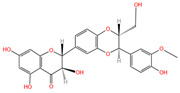
5-Hydroxy-6,6-dimethylpyrano flavone	7.95 × 10^6^	0.994	320.10509	C_20_H_16_O_4_	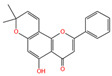

**Table 4 foods-12-04256-t004:** Variance and significant analysis of the Box–Behnken design test.

Source	Sum of Squares	df	Mean Square	*F*-Value	*p*-Value
Model	1482.84	9	164.76	555.31	<0.0001
*A*	2.11	1	2.11	7.12	0.0321
*B*	13.13	1	13.13	44.26	0.0003
*C*	74.42	1	74.42	250.83	<0.0001
*AB*	1.7	1	1.70	5.74	0.0478
*AC*	1.04	1	1.04	3.51	0.1033
*BC*	64.16	1	64.16	216.25	<0.0001
*A^2^*	300.80	1	300.8	1013.82	<0.0001
*B^2^*	369.85	1	369.85	1246.54	<0.0001
*C^2^*	518.29	1	518.29	1746.84	<0.0001
Residual	2.08	7	0.30		
Lack of Fit	0.19	3	0.062	0.13	0.9364
Pure Error	1.89	4	0.47		
Cor Total	1484.92	16			

## Data Availability

The datasets generated for this study are available on request to the corresponding author.

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
