# Peer review of "Antioxidant Activity, Formulation, Optimization and Characterization of an Oil-in-Water Nanoemulsion Loaded with Lingonberry (Vaccinium vitis-idaea L.) Leaves Polyphenol Extract"

_foods, 2023, doi:10.3390/foods12234256_

Round 1

Reviewer 1 Report

Comments and Suggestions for Authors

The authors reported article: Antioxidant Activity, Formulation, Optimization, Characterization and Stability of the oil in water Nanoemulsion loaded Lingonberry (Vaccinium vitis-idaea L.) leaves polyphenol Extract

The subject is very topical, it is no secret that we live in a polluted world and efforts to improve this situation are vital.

The data presented in this paper is important. The study showed that combining different methods together, it is possible to analyze obtained extracts from different sides, also antioxidant activity is determined.

The authors have well reviewed the use of  Lingonberry (Vaccinium vitis-idaea L.) leaves. About the chemical composition (the main groups of chemical compounds) that have a known positive effect on the cells of living organisms, their protection against various diseases.

Innovative research methods for comprehensive profiling of chemical compounds have been demonstrated.

Innovative research methods for comprehensive profiling of chemical compounds have been demonstrated.

Why was only one extraction approach used? Perhaps with other extraction solvents it is possible to obtain extracts with higher yields and more dominant target compounds?

Author Response

Comments 1: Why was only one extraction approach used? Perhaps with other extraction solvents it is possible to obtain extracts with higher yields and more dominant target compounds?

Response 1: [Thank you for pointing this out. I/We agree with this comment. Therefore, we have explained the reasons for using this extraction method in page 4 paragraph lines 133-136 of the revised manuscript, with the following modifications: ]

“[Taking into account edible safety and the foundation established by our research team[19], we employed food-grade ethanol for the solvent extraction of polyphenols. The extraction procedure was adapted from the methodology presented in[20], with minor modifications.] 

Reviewer 2 Report

Comments and Suggestions for Authors

In the present paper authors describe antioxidant activity, formulation, optimization and characterization of the oil in water nanoemulsion loaded Lingonberry leaves polyphenol extract.

Technically there are many problems all the way through the text:

1) All variables need to be put to italic through the whole manuscript.

2) You should pay more attention to the space between the variable, the numerical value and the unit of measurement throughout the text (line 119, line 184, line 302…).

3) In the text, Figure 2 is listed instead of Figure 3 (line 278).

4) In the text, Figure 3 is listed instead of Figure 4 (line 282, line 283).

5) In the text, Figure 4 is listed instead of Figure 5 (line 303).

6) In the text, Figure 5 is listed instead of Figure 6 (line 309).

7) In Table 2, the measurement units for the variables are not specified.

8) The manufacturers of the measuring devices that were used in the measurements are not listed in experimental section.

9) There are also many typographical errors through the whole manuscript (line 46, line 176, line 226, line 271…).

10) The value of the zeta potential should be negative (line 310).

Overall merit is very low, paper is badly written, it lacks novelty and important experimental information, so I do not recommend this paper to be published in this journal.

Author Response

Comments 1: All variables need to be put to italic through the whole manuscript.

Response 1: [Thank you for pointing this out. I/We agree with this comment. Therefore, we have modified all variables in the manuscript to italicize.Modified variables in page 4 line 172 173; page 5 line 176 177 179 180; page 10 line 306-308; page 11 line 310-312 314 326 327 329-333 of the revised manuscript.]

Comments 2: You should pay more attention to the space between the variable, the numerical value and the unit of measurement throughout the text (line 119, line 184, line 302…).

Response 2: [Thank you for pointing this out. I/We agree with this comment. Therefore, we have modified the space relationship between variables, numerical values, and measurement units in the manuscript..The changes can be found on the page 4 line 138;  page 6 line 214;  page 12 line 344;  page 14 line 381 382 of the revised manuscript.]

“[ page 4 line 138: freeze-stored in -20 â„ƒ refrigerator and sealed for storage.

page 6 line 214: 40 magnification at 20 ℃

page 12 line 344:nanoemulsion with 114.52 ± 0.015 nm

page 14 line 381 382:The percentage inhibition obtained was maximum for ascorbic acid (85.43% ± 2.85%) compared to the LBPN (79.37% ± 3.18%) and LBP (63.58% ± 2.95%). ]”

Comments 3: In the text, Figure 2 is listed instead of Figure 3 (line 278).

Response 3:Thank you for pointing this out. I/We agree with this comment.Therefore, we have corrected Fig. 2 to Fig. 3.This change is located in the line 35-64 69-119.

[page 11 line 319:Fig 3. illustrates the normal probability plot residuals for the entrapping efficiency.] 

Comments 4:  In the text, Figure 3 is listed instead of Figure 4 (line 282, line 283).

Response 4:Thank you for pointing this out. I/We agree with this comment.Therefore, we have corrected Fig. 3 to Fig. 4.This change is located in the page 11 line 324.

[page 11 line 324:Fig. 4 wherein two factors are held...]

Comments 5:  In the text, Figure 4 is listed instead of Figure 5 (line 303).

Response 5:Thank you for pointing this out. I/We agree with this comment.Therefore, we have corrected Fig. 4 to Fig. 5.This change is located in the page 12 line 345.

[ page 12 line 345.:As depicted in Fig. 5...]

Comments 6:In the text, Figure 5 is listed instead of Figure 6 (line 309).

Response 6:Thank you for pointing this out. I/We agree with this comment.Therefore, we have corrected Fig. 4 to Fig. 5.This change is located in the page 13 line 352.

[page 13 line 352.:Fig. 6 reveals that ...]

Comments 7: In Table 2, the measurement units for the variables are not specified.

Response 7:Thank you for pointing this out. I/We agree with this comment.Therefore, we added units for each variable.This change is located in the page 4 line 172.

[page 4 line 172:

Protein concentration (X1,%)

Oil phase mass fraction (X2,%)

Extract mass fraction (X3,%)

...]

Comments 8:  The manufacturers of the measuring devices that were used in the measurements are not listed in experimental section.

Response 8:Thank you for pointing this out. I/We agree with this comment. Therefore, we have added manufacturers of all instruments used in the text.This change is located in the page 3 line 131;  page 4 line 137; page 4 line 140; page 4 line 148; page 4 line 157; page 5 line 202; page 6 line 209; page 6 line 214.

[page 3 line 131:Subsequently, the samples were lyophilized using a freeze dryer (SP Scientific, Gardiner, USA).

page 4 line 137:The freeze-dried Lingonberry leaves were first pulverized into a fine powder using FW-100 high-speed universal grinder (Braun, Germany).

page 4 line 140: then homogenize (BME 100L, Weiyu, China) with a high shear for 3 min.

page 4 line 148: UPLC-TQ-MS was performed on a Waters ACQUITY UPLCTM system (Waters, Milford, USA).

page 4 line 157: Ultrasonic homogenizer (ASU-10D, AS one, Japan) was used to produce nanoemulsions.

page 5 line 202.: Nano-ZS PALS laser particle size analyzer (Malvern, UK).

Page 6 line 209: Nano-ZS PALS laser particle size analyzer (Malvern, UK).

Page 6 line 214: confocal laser scanning microscope (Leica, Germany)]

Comments 9: There are also many typographical errors through the whole manuscript (line 46, line 176, line 226, line 271…).

Response 9:Thank you for pointing this out. I/We agree with this comment.Therefore, we have corrected all typographical errors in the manuscript. This change is located in the page 2 line 52; page 6 line 205; page 7 line 258; page 11 line 314  .

[page 2 line 52.: organic acids, carbohydrates, glycosyl hydroquinones, and proanthocyanidins[4].

page 6 line 205: temperature of 25 ℃ ± 0.1 ℃.

Page 7 line 258: Apigenin 7-rhamnosyl-(1->2)-galacturonide, Bis(4-ethylbenzylidene)sorbitol

Page 11 line314: where A is the protein concentration; B is the oil phase mass fraction; C is the extract mass fraction.]

Comments 10:  The value of the zeta potential should be negative (line 310).

Response 10:Thank you for pointing this out. I/We agree with this comment.Therefore, we have corrected the incorrect values in the manuscript.This change is located in the page 13 line 353.

[page 13 line 353.:a zeta potential of -23.29 ± 0.4471 mV. ]

Reviewer 3 Report

Comments and Suggestions for Authors

This manuscript entitled “Antioxidant Activity, Formulation, Optimization and Characterization of the oil in water Nanoemulsion loaded gonberry (Vaccinium vitis-idaea L.) leaves polyphenol Extract” is interesting; however, the manuscript needs some modification:

1.      In the line 107 (for SPI, WPI, and MCT) and in the entire text (line 121: LBP; ,when an abbreviation is used for the first time, its full word must be mentioned in the text.

2.      In the line 115, the freeze-drying process specifications and device model must be mentioned.

3.      In the line 118 and 119, the revolutions/rotations per minute for stirrer and the model of homogenizer be added in the text.

4.      In line 121, 2.3. Characterization of LBP by UPLC-TQ-MS section; if a procedure for sample preparation was performed before injection into the column, it should be added with details in the text.

5.      In formulas 2 and 3, the words sample and control should be put in lowercase e.g.  A sample.

6.      In result, table 3 and figure 3 should better placed after the result of 3.1. UPLC-TQ-MS Characterization of Extract section.

7.      If possible, the efficiency of the extract obtained from the dried leaves of the plant should be reported in the results section.

8.      The numbers and titles of the horizontal and vertical axes of Figure 3, Figure 4, and figure 8, should be written in a larger font or images with a higher resolution should be used.

Author Response

Comments 1: In the line 107(for SPI, WPI, and MCT) and in the entire text (line 121: LBP; ,when an abbreviation is used for the first time, its full word must be mentioned in the text.

Response 1: Thank you for pointing this out. I/We agree with this comment. Therefore, we have added freeze-drying equipment and processes in the manuscript. This change is located in the page 3 line 130 131.

[It consists of 10% (w/w) lingonberry extract and 20% (w/w) medium chain triglyceride (MCT). Additionally, a surfactant mixture was used, which combined soy protein isolate (SPI) and whey protein isolate (WPI) at 4% (w/w).

The antioxidant activity (DPPH inhibition rate, ABTS inhibition rate and total reducing power) of lingonberry nanoemulsion was significantly higher than that of the lingonberry polyphenols (LBP) extract.] 

Comments 2: In the line 115, the freeze-drying process specifications and device model must be mentioned.

Response 2: Thank you for pointing this out. I/We agree with this comment. Therefore, we have provided the full name in the abstract section. This change is located in the page 1 line 23 24 27.

[ Subsequently, the samples were lyophilized using a freeze dryer (SP Scientific, Gardiner, USA).

] 

Comments 3: In the line 118 and 119, the revolutions/rotations per minute for stirrer and the model of homogenizer be added in the text.

Response 3:Thank you for pointing this out. I/We agree with this comment. Therefore, we have added revolutions per minute and homogenizer model in the manuscript. This change is located in the page 4 line 140.

[Add 400 mL of 75% pre-cooled ethanol and mix with RZ-8012 handheld stirrer at 12000 r/m for 5 min, then homogenize (BME 100L, Weiyu, China) with a high shear for 3 min. ] 

Comments 4:  In line 121, 2.3. Characterization of LBP by UPLC-TQ-MS section; if a procedure for sample preparation was performed before injection into the column, it should be added with details in the text.

Response 4:Thank you for pointing this out. I/We agree with this comment. Therefore, we have supplemented the sample preparation process before UPLC-TQ-MS in the manuscript. This change is located in the page 4 line 143-146.

[The extract of lingonberry leaves, prepared as described in section 2.2, was evaporated to dryness using a rotary evaporator at 45 ℃ under vacuum conditions. The resulting extract was then redissolved in 5 mL of methanol and subsequently filtered through a 0.45 μm filter prior to analysis. ] 

Comments 5:  In formulas 2 and 3, the words sample and control should be put in lowercase e.g. A sample.

Response 5:Thank you for pointing this out. I/We agree with this comment. Therefore, we have corrected the writing issue in the formula.. This change is located in the page 6 line 229 238.

[

DPPH radical scavenging activity(%)= (1-Asample/Acontrol)×100

(2)

ABTS radical scavenging activity(%)= (1-Asample/Acontrol)×100

(3)

Comments 6:In result, table 3 and figure 3 should better placed after the result of 3.1. UPLC-TQ-MS Characterization of Extract section.

Response 6:Thank you for pointing this out. I/We agree with this comment. Therefore, we adjusted the position of Table 3 in Figure 3. This change is located in the page 8 line 299-301.

Comments 7: If possible, the efficiency of the extract obtained from the dried leaves of the plant should be reported in the results section.

Response 7:Thank you for pointing this out. I/We agree with this comment. Therefore, we added the polyphenol extraction rate to the results. This change is located in the page 7 line 254 255.

[The polyphenol extraction rate was calculated based on the method outlined in[25] and was found to be 138.18 mg/g. ] 

Comments 8: The numbers and titles of the horizontal and vertical axes of Figure 3, Figure 4, and figure 8, should be written in a larger font or images with a higher resolution should be used.

Response 8:Thank you for pointing this out. I/We agree with this comment. Therefore, we have increased the font size in Figures 3, 4, and 8. This change is located in the page 9 line 303; page 10 line 305; page 13 line 372.

Reviewer 4 Report

Comments and Suggestions for Authors

The authors studied "Antioxidant Activity, Formulation, Optimization, and Charac-2 Terization of the Oil in Water Nanoemulsion-Lin-3 Gonberry (Vaccinium vitis-idaea L.) Leaves Polyphenol Extract". The authors investigate the effects of different formulation variables on the stability and antioxidant activity of the nanoemulsion, as well as characterizing the polyphenolic compounds present in the lingonberry extract using UPLC-TQ-MS. Overall, the work is interesting, and I find the manuscript informative. However, there are some revisions and modifications to improve the quality of the work. I will recommend it after some revisions.

1.     The abstract could benefit from a clearer statement of the research question and the main findings of the study. The abstract should be concise but informative, and it should provide a clear overview of the research.

2.     Keywords are not consistent. To ensure consistency, authors can choose one style, either all lowercase or all uppercase, for the keywords.

3.     The introduction could be expanded to provide more context for the study and explain why the research is important. The introduction should provide a clear rationale for the research and explain why it is needed.

4.     What about the potential toxicity of the nanoemulsion loaded with Lingonberry leaf polyphenol extract, and how might this affect the human body?

5.     What are the health-promoting effects of lingonberry on the human body? How does the nanoemulsion delivery system enhance the stability and antioxidant activity of lingonberry leaf extract? What are the other bioactive compounds found in Lingonberry leaf polyphenol extract besides apigenin, sorbitol, and hesperidin?

6.     Conclusions should not end abruptly. Authors should conclude by summarizing the significance of their findings and how they contribute to the field.

7.     The article could benefit from a more detailed discussion of the implications of the findings for future research or practical applications. The article should provide more information on how the research could be applied in practice and should discuss the potential benefits and limitations of the research.

Comments on the Quality of English Language

There are many confusing statements in the manuscript. The authors must rewrite many sentences for better understanding

Author Response

Comments 1:  The abstract could benefit from a clearer statement of the research question and the main findings of the study. The abstract should be concise but informative, and it should provide a clear overview of the research.

Response 1: Thank you for pointing this out. I/We agree with this comment. Therefore, we have revised the abstract section, removed unnecessary parts, and made the results more clear.This change is located in the page 1 line 14-30.

[The active ingredients in lingonberry leaves and their beneficial properties to the human body have been well confirmed. In order to improve the stability and antioxidant activity of the active ingredients in lingonberry leaves, the response surface optimization method was used to prepare the oil in water nanoemulsion of polyphenol extract from lingonberry leaves. The active components in the extract were analyzed by ultra-performance liquid chromatography-triple quadrupole-mass spectrometry (UPLC-TQ-MS), and the bioactive compounds such as apigenin, sorbitol and hesperidin were mainly found. Nanoemulsion droplets of 120 nm in diameter was prepared by ultrasonic emulsification. The optimal nanoemulsion formulation was determined through rigorous testing. It consists of 10% (w/w) lingonberry extract and 20% (w/w) medium chain triglyceride (MCT). Additionally, a surfactant mixture was used, which combined soy protein isolate (SPI) and whey protein isolate (WPI) at 4% (w/w). The preparation method utilized ultrasonic emulsification,  applying an ultrasonic power of 360W for a duration of 300 s. The antioxidant activity (DPPH inhibition rate, ABTS inhibition rate and total reducing power) of lingonberry nanoemulsion was significantly higher than that of the lingonberry polyphenols (LBP) extract. The nanoemulsion prepared by the optimal formulation had an entrapping efficiency of 73.25% ± 0.73% and a diameter of 114.52 ± 0.015 nm, with a satisfactory particle size of nanoscale and a PDI of 0.119 ± 0.065, demonstrating the good stability of the emulsions. ] 

Comments 2: Keywords are not consistent. To ensure consistency, authors can choose one style, either all lowercase or all uppercase, for the keywords.

Response 2: Thank you for pointing this out. I/We agree with this comment. Therefore, we have revised the keywords to ensure that all keywords are lowercase. This change is located in the page 1 line 31 32.

[Keywords: nanoemulsion; response surface methodology; antioxidant activity; lingonberry (Vaccinium vitis-idaea L.); delivery system] 

Comments 3: The introduction could be expanded to provide more context for the study and explain why the research is important. The introduction should provide a clear rationale for the research and explain why it is needed.

Response 3:Thank you for pointing this out. I/We agree with this comment. Therefore, we have enriched the content of the introduction section and added recent research progress to explain the research background and reasons.This change is located in the page 1 line 31 32.

[The lingonberry (Vaccinium vitis-idaea L., family Ericaceae) is a natural wild berry with significant nutritional value and a wide range of health-promoting properties. The lingonberry is rich in phenolics, which are the main active substances in this fruit. ingonberry, often termed a "superfruit" is renowned for its substantial antioxidant properties. Predominantly found in regions of northern and central Europe, Canada, and Asia—including mountainous zones like North-eastern China's Greater Khingan and Lesser Hinggan Ranges—it's a source of various potent bioactives[1]. While bilberry and lingonberry leaves emerge as primary by-products during berry collection, recent studies have highlighted an intriguing observation[2]. These leaves and stems from the Vaccinium species house a more substantial phenolic content than the berries themselves. This elevated phenolic concentration is congruent with their superior antioxidant capabilities compared to the fruit[3]. Such findings suggest that these by-products could be harnessed as promising reservoirs of bioactive compounds, opening doors for their inclusion in food supplements, nutraceuticals, or functional food products.

Diving deeper into the biochemical makeup of lingonberry leaves, they feature a plethora of compounds, including phenolic acids, anthocyanins, triterpene acids, organic acids, carbohydrates, glycosyl hydroquinones, and proanthocyanidins[4]. This rich profile renders the leaves a more chemically diverse entity than the berry fruits. Notably, recent research has spotlighted flavonoid glycosides as the dominant phenolic entities in lingonberries. These compounds exhibit a myriad of health-promoting attributes, such as astringent, antitussive, diuretic, neuroprotective, antioxidant, and anti-inflammatory activities. Their potential roles in inhibiting cancer cell proliferation further underscore their therapeutic potential[5].

Beyond these properties, there are studies indicating that extracts from lingonberry leaves can curtail hepatitis C virus expression and stymie the proliferation of sensitive human promyelocytic leukaemia HL60 cells. Historically and traditionally, these leaves have been employed to mitigate cholesterol, address gastric complications, treat rheumatic ailments, and combat infections in the bladder and kidneys[6-8].

Several studies have successfully constructed the delivery system of polyphenol nanoemulsion, improving the stability[10,11], biological accessibility[12-14], antioxidant[15,16] and other functional activities of polyphenols. Nanoemulsions have been widely used to improve the bioavailability of insoluble drugs.

It's imperative to note that nanoemulsions aren't equilibrium systems. They are non-thermodynamically stable and don't form spontaneously. Their formation is influenced either by their inherent chemical potential within the emulsion system or by external forces. Emulsification methods, pivotal to their formation, can be broadly classified based on the energy source utilized. There are high-energy methods, relying on external energy, and low-energy methods that harness the system's inherent potential.

Among the high-energy methods, techniques like ultrasound, micronization, and high-pressure homogenization are prevalent. Notably, ultrasound technology stands out due to its efficiency and cost-effectiveness. It delivers nanoemulsions with a smaller droplet size, a reduced polydispersity index (PDI), and superior stability, all while using fewer surfactants. The technology employs a frequency of at least 20 kHz[17]. The ultrasonic vibrations produce pronounced shear and pressure gradients in the emulsion. This facilitates the breakdown of larger droplets, enabling the continuous production of nanoemulsions. Notably, the energy consumption and associated costs are considerably lower than those in high-pressure homogenization.

To conclude, as the quest for optimal drug delivery systems persists, nanoemulsions are progressively recognized for their multifaceted benefits, particularly in the pharmaceutical and functional food industries. Their potential in enhancing the delivery and efficacy of compounds like LBP is undeniable[18].] 

Comments 4:  What about the potential toxicity of the nanoemulsion loaded with Lingonberry leaf polyphenol extract, and how might this affect the human body?

Response 4:Thank you for pointing this out. I/We agree with this comment. Therefore, we discussed the potential toxicity of LBPN in the conclusion section. This change is located in the page 14 line 233-245.

[Nanocarriers have been primarily used in drug delivery systems and the development of functional foods. It is therefore essential to delve deeper into studies concerning their toxicity and underlying mechanisms of any toxic effects, in order to determine the permissible daily intake of nanoemulsions.The absorption and metabolic processes of nanoscale carriers differ from those of microscale carriers. Currently, the potential toxicity of nanoscale emulsions upon absorption by the human body remains unclear and requires further in-depth investigation, such as a systematic evaluation of the potential risks of LBPN through toxicological experiments, including clinical studies.] 

Comments 5:  What are the health-promoting effects of lingonberry on the human body? How does the nanoemulsion delivery system enhance the stability and antioxidant activity of lingonberry leaf extract? What are the other bioactive compounds found in Lingonberry leaf polyphenol extract besides apigenin, sorbitol, and hesperidin?

Response 5:Thank you for pointing this out. I/We agree with this comment. Therefore, we supplemented the health promotion effect of Lingonberry in the introduction (This change is located in the page 2 line 59-63.), and discussed how the nanoemulsion system can improve the stability and activity of the embedded materials by introducing references in similar fields in the result part.(This change is located in the page 14 line 384-390). In the results section, additional explanations were provided on the other active ingredients in the extract. This change is located in the page 7 line 294-298.

[page 2 line 59-63: Beyond these properties, there are studies indicating that extracts from lingonberry leaves can curtail hepatitis C virus expression and stymie the proliferation of sensitive human promyelocytic leukaemia HL60 cells. Historically and traditionally, these leaves have been employed to mitigate cholesterol, address gastric complications, treat rheumatic ailments, and combat infections in the bladder and kidneys[6-8].

page 14 line 384-390: Research has shown that the preparation of resveratrol as a resveratrol nanoscale emulsion resulted in a DPPH inhibition rate as high as 83.93±3.81%, an increase of nearly 13% compared to non-encapsulated resveratrol solutions[15]. Kumar etal[11] employed lecithin and Tween-80 as emulsifiers to prepare a resveratrol-loaded nanoscale emulsion, which remained stable without phase separation even after four months of storage. Under ultraviolet radiation exposure, the nanoscale emulsion significantly slowed down and inhibited the degradation of resveratrol.

page 7 line 294-298: In addition to the aforementioned active substances, the extract also contains a relatively abundant variety of phenolic compounds, including quercetin, oleanolic acid and various other phenolic compounds, totaling 14 different types. These findings are in line with previous research on the polyphenolic content of lingonberry leaves[20,33].] 

Comments 6:Conclusions should not end abruptly. Authors should conclude by summarizing the significance of their findings and how they contribute to the field.

Response 6:Thank you for pointing this out. I/We agree with this comment. Therefore, we have added the significance of our research and contributions to this field in the conclusion section of the manuscript.This change is located in the page 14 line 407-418.

[In conclusion, this study successfully developed a lingonberry polyphenol extract-loaded oil-in-water nanoemulsion using a response surface optimization method. The nanoemulsion exhibited small droplet size, good stability, and enhanced antioxidant activity compared to the extract. These findings contribute to the understanding of the formulation and optimization of nanoemulsions for the delivery of bioactive compounds.

Moving forward, further research can explore the potential applications of this lingonberry nanoemulsion in the food and pharmaceutical industries. The optimized formulation can be utilized to enhance the stability and bioavailability of bioactive compounds, leading to improved functional food products and nutraceuticals. Additionally, the study can be extended to investigate the in vivo efficacy and safety of the nanoemulsion, providing valuable insights for future clinical applications. ] 

Comments 7:The article could benefit from a more detailed discussion of the implications of the findings for future research or practical applications. The article should provide more information on how the research could be applied in practice and should discuss the potential benefits and limitations of the research.

Response 7:Thank you for pointing this out. I/We agree with this comment. Therefore, in the conclusion section of the manuscript, we discussed in more detail the significance of the research content for practical applications. This change is located in the page 14 line 419-438.

[Furthermore, it is important to consider the practical implications of this research. The development of a stable and bioactive nanoemulsion opens up opportunities for the utilization of lingonberry polyphenols in various products, such as functional beverages, dietary supplements, and topical formulations. However, it is crucial to address the potential limitations and challenges associated with the scale-up and commercialization of the nanoemulsion, including cost-effectiveness, regulatory considerations, and consumer acceptance. Nanocarriers have been primarily used in drug delivery systems and the development of functional foods. It is therefore essential to delve deeper into studies concerning their toxicity and underlying mechanisms of any toxic effects, in order to determine the permissible daily intake of nanoemulsions.The absorption and metabolic processes of nanoscale carriers differ from those of microscale carriers. Currently, the potential toxicity of nanoscale emulsions upon absorption by the human body remains unclear and requires further in-depth investigation, such as a systematic evaluation of the potential risks of LBPN through toxicological experiments, including clinical studies.

In summary, this study lays the foundation for the application of lingonberry polyphenols in nanoemulsion formulations, highlighting their potential for improving stability, bioavailability, and antioxidant activity. Further research and development efforts are warranted to fully explore the practical applications and benefits of this technology in the food and pharmaceutical industries.] 

Reviewer 5 Report

Comments and Suggestions for Authors

Dear authors,

I am providing suggestions for improving your manuscript:

Abstract:

• Please shorten the abstract and do not reference literature within it.

• In the first paragraph, the sentence "The active ingredients in Lingonberry leaves and its beneficial properties to human body" contains a grammatical error. The correct form is "The active ingredients in Lingonberry leaves and their beneficial properties to the human body."

• In the sentence "The Lingonberry is rich in phenolics, which are the main active substances in the Lingonberry," the word "Lingonberry" is used twice in one sentence. You can improve it to "The Lingonberry is rich in phenolics, which are the main active substances in this fruit," to avoid repetition.

• In the third paragraph, the sentence "The optimal nanoemulsion formulation is 10% (w/w) of Lingonberry extract, 20% (w/w) of medium chain triglyceride (MCT) and surfactant mixture of soy protein isolate (SPI) and whey protein isolate (WPI) (4%, w/w) prepared by ultrasonic emulsification (ultrasonic power 360W) for 300s" is long and hard to read. I suggest breaking this sentence into more understandable parts to improve clarity.

• In the sentence "The antioxidant activity (DPPH inhibition rate, ABTS inhibition rate, and total reducing power) of Lingonberry nano emulsion was significantly higher than that of the non-embedded extract," you can explain what "non-embedded extract" means to avoid ambiguity.

Introduction:

• In the sentence "The Lingonberry plant is an evergreen terrestrial shrub belonging to the genus Echinacea," there is a factual error. Lingonberry is not a species of the Echinacea plant but a separate species (Vaccinium vitis-idaea). This should be corrected to avoid conveying false information.

• In the sentence "The Lingonberry is a natural wild berry with high nutritional value of a wide spectrum of health-promoting effects on wide-ranging diseases," it seems somewhat lengthy and imprecise. It can be expressed more concisely, for example: "Lingonberry is a natural wild berry with significant nutritional value and a wide range of health-promoting properties."

• In the other sentences in this section, I did not find any language or substantive errors. However, the authors may consider presenting some information in a more concise and clear manner to facilitate reader understanding.

• There are too few literature references in the introduction. Please add the latest literature findings. Currently, the literature review is very weak.

Materials and Methods:

• In the sentence "The plant material was acquired from kudu Forestry Bureau, Hulunbeir, Inner Mongolia Autonomous Region," there may be a lack of specificity, so it's worth adding information about the type of plant (lingonberry) and its significance for the research. Furthermore, "kudu Forestry Bureau" should be capitalized as "Kudu Forestry Bureau."

• In the sentence "The plant material was then transported to the laboratory, cleaned the residual soil of the Lingonberry and separated them according to their parts," there is likely a sentence structure error. It could be corrected to: "The plant material was then transported to the laboratory, where it was cleaned of residual soil, and the different parts were separated."

• In the sentence "sealed them in a sealed container," the word "sealed" is used redundantly. You can shorten it to "placed in a container" or something similar.

• In the sentence "The freeze-dried Lingonberry leaves were firstly pulverized into fine powder," there is a small grammatical error. Correctly: "The freeze-dried Lingonberry leaves were first pulverized into a fine powder."

• In the sentence "The positive mobile phase was prepared from 0.1% FA-H2O (solvent A) and MeOH (solvent B), and the negative was prepared from 5 mM Ammonium acetate-H2O (solvent A:Ammonia water regulation, pH=9) and MeOH (solvent B), with a flow rate of 0.35 mL/min," there is a minor ambiguity in the sentence structure. You can clarify this by expressing it more understandably, for example: "The positive mobile phase (solvent A) was prepared using 0.1% formic acid (FA) in water and MeOH, while the negative phase (solvent B) was created using 5 mM Ammonium acetate in water (Ammonia water regulation, pH=9) and MeOH. The flow rate was set to 0.35 mL/min."

In the sentence "The mean droplet diameter, poly dispersity index (PDI), were measured," there is a plural error. Correctly: "The mean droplet diameter and polydisperse index (PDI) were measured."

Results:

The results should be analyzed in relation to the results of other researchers conducting research in the same or very similar areas. Please add this.

Conclusions:

The conclusions in this fragment appear to be consistent with the results presented earlier in the article, and I do not notice any visible errors in this context.

Comments on the Quality of English Language

All the comments regarding language aspects have also been included in the previous section. The language needs to be checked as there are many errors.

Author Response

Comments 1: Please shorten the abstract and do not reference literature within it.

Response 1:Thank you for pointing this out. I/We agree with this comment.Therefore, we have streamlined the abstract section and removed references. This change is located in the page 1 line 14-30.

[The active ingredients in lingonberry leaves and their beneficial properties to the human body have been well confirmed. In order to improve the stability and antioxidant activity of the active ingredients in lingonberry leaves, the response surface optimization method was used to prepare the oil in water nanoemulsion of polyphenol extract from lingonberry leaves. The active components in the extract were analyzed by ultra-performance liquid chromatography-triple quadrupole-mass spectrometry (UPLC-TQ-MS), and the bioactive compounds such as apigenin, sorbitol and hesperidin were mainly found. Nanoemulsion droplets of 120 nm in diameter was prepared by ultrasonic emulsification. The optimal nanoemulsion formulation was determined through rigorous testing. It consists of 10% (w/w) lingonberry extract and 20% (w/w) medium chain triglyceride (MCT). Additionally, a surfactant mixture was used, which combined soy protein isolate (SPI) and whey protein isolate (WPI) at 4% (w/w). The preparation method utilized ultrasonic emulsification,  applying an ultrasonic power of 360W for a duration of 300 s. The antioxidant activity (DPPH inhibition rate, ABTS inhibition rate and total reducing power) of lingonberry nanoemulsion was significantly higher than that of the lingonberry polyphenols (LBP) extract. The nanoemulsion prepared by the optimal formulation had an entrapping efficiency of 73.25% ± 0.73% and a diameter of 114.52 ± 0.015 nm, with a satisfactory particle size of nanoscale and a PDI of 0.119 ± 0.065, demonstrating the good stability of the emulsions. ]

Comments 2: In the first paragraph, the sentence "The active ingredients in Lingonberry leaves and its beneficial properties to human body" contains a grammatical error. The correct form is "The active ingredients in Lingonberry leaves and their beneficial properties to the human body."

Response 2: Thank you for pointing this out. I/We agree with this comment.Therefore, we have revised the wording of the statement. This change is located in the page 1 line 14.

[The active ingredients in lingonberry leaves and their beneficial properties to the human body.] 

Comments 3:In the sentence "The Lingonberry is rich in phenolics, which are the main active substances in the Lingonberry," the word "Lingonberry" is used twice in one sentence. You can improve it to "The Lingonberry is rich in phenolics, which are the main active substances in this fruit," to avoid repetition.

Response 3: Thank you for pointing this out. I/We agree with this comment.Therefore, we have revised the wording of the statement. This change is located in the page 1 line 36.

[The lingonberry is rich in phenolics, which are the main active substances in this fruit. ] 

Comments 4: In the third paragraph, the sentence "The optimal nanoemulsion formulation is 10% (w/w) of Lingonberry extract, 20% (w/w) of medium chain triglyceride (MCT) and surfactant mixture of soy protein isolate (SPI) and whey protein isolate (WPI) (4%, w/w) prepared by ultrasonic emulsification (ultrasonic power 360W) for 300s" is long and hard to read. I suggest breaking this sentence into more understandable parts to improve clarity.

Response 4:Thank you for pointing this out. I/We agree with this comment.Therefore, we have revised the wording of the statement. This change is located in the page 1 line 21-25.

[The optimal nanoemulsion formulation was determined through rigorous testing. It consists of 10% (w/w) lingonberry extract and 20% (w/w) medium chain triglyceride (MCT). Additionally, a surfactant mixture was used, which combined soy protein isolate (SPI) and whey protein isolate (WPI) at 4% (w/w). The preparation method utilized ultrasonic emulsification,  applying an ultrasonic power of 360W for a duration of 300 s.]

Comments 5: In the sentence "The antioxidant activity (DPPH inhibition rate, ABTS inhibition rate, and total reducing power) of Lingonberry nano emulsion was significantly higher than that of the non-embedded extract," you can explain what "non-embedded extract" means to avoid ambiguity.

Response 5:Thank you for pointing this out. I/We agree with this comment.Therefore, we have revised the wording of the statement. This change is located in the page 1 line 27 28.

[The antioxidant activity (DPPH inhibition rate, ABTS inhibition rate and total reducing power) of lingonberry nanoemulsion was significantly higher than that of the lingonberry polyphenols (LBP) extract.] 

Comments 6: In the sentence "The Lingonberry plant is an evergreen terrestrial shrub belonging to the genus Echinacea," there is a factual error. Lingonberry is not a species of the Echinacea plant but a separate species (Vaccinium vitis-idaea). This should be corrected to avoid conveying false information.

Response 6:Thank you for pointing this out. I/We agree with this comment.Therefore, we have revised the wording of the statement. This change is located in the page 1 line 35 36.

[The lingonberry (Vaccinium vitis-idaea L., family Ericaceae) is a natural wild berry with significant nutritional value and a wide range of health-promoting properties.] 

Comments 7:  In the sentence "The Lingonberry is a natural wild berry with high nutritional value of a wide spectrum of health-promoting effects on wide-ranging diseases," it seems somewhat lengthy and imprecise. It can be expressed more concisely, for example: "Lingonberry is a natural wild berry with significant nutritional value and a wide range of health-promoting properties."

Response 7:Thank you for pointing this out. I/We agree with this comment.Therefore, we have revised the wording of the statement. This change is located in the page 1 line 35 36.

[The lingonberry (Vaccinium vitis-idaea L., family Ericaceae) is a natural wild berry with significant nutritional value and a wide range of health-promoting properties.] 

Comments 8: In the other sentences in this section, I did not find any language or substantive errors. However, the authors may consider presenting some information in a more concise and clear manner to facilitate reader understanding.

Response 8:Thank you for pointing this out. I/We agree with this comment.Therefore, we have revised the wording of the statement. This change is located in the page 1 line 35 36.

[The lingonberry (Vaccinium vitis-idaea L., family Ericaceae) is a natural wild berry with significant nutritional value and a wide range of health-promoting properties.] 

Comments 9: There are too few literature references in the introduction. Please add the latest literature findings. Currently, the literature review is very weak.

Response 9:Thank you for pointing this out. I/We agree with this comment.Therefore, we have re optimized the wording of the entire manuscript.

Comments 10:  In the sentence "The plant material was acquired from kudu Forestry Bureau, Hulunbeir, Inner Mongolia Autonomous Region," there may be a lack of specificity, so it's worth adding information about the type of plant (lingonberry) and its significance for the research. Furthermore, "kudu Forestry Bureau" should be capitalized as "Kudu Forestry Bureau."

Response 10:Thank you for pointing this out. I/We agree with this comment.Therefore, we have revised the wording of the statement. This change is located in the page 3 line 126.

[The plant material was acquired from Kudu Forestry Bureau, Hulunbeir, Inner Mongolia Autonomous Region.] 

Comments 11:  In the sentence "The plant material was then transported to the laboratory, cleaned the residual soil of the Lingonberry and separated them according to their parts," there is likely a sentence structure error. It could be corrected to: "The plant material was then transported to the laboratory, where it was cleaned of residual soil, and the different parts were separated."

Response 11:Thank you for pointing this out. I/We agree with this comment.Therefore, we have revised the wording of the statement. This change is located in the page 3 line 128 129.

[The plant material was then transported to the laboratory, where it was cleaned of residual soil, and the different parts were separated, ] 

Comments 12: In the sentence "sealed them in a sealed container," the word "sealed" is used redundantly. You can shorten it to "placed in a container" or something similar.

Response 12:Thank you for pointing this out. I/We agree with this comment.Therefore, we have revised the wording of the statement. This change is located in the page 3 line 130.

[placed in a container] 

Comments 13:In the sentence "The freeze-dried Lingonberry leaves were firstly pulverized into fine powder," there is a small grammatical error. Correctly: "The freeze-dried Lingonberry leaves were first pulverized into a fine powder."

Response 13:Thank you for pointing this out. I/We agree with this comment.Therefore, we have revised the wording of the statement. This change is located in the page 4 line 136 137.

[The freeze-dried Lingonberry leaves were first pulverized into a fine powder] 

Comments 14:In the sentence "The positive mobile phase was prepared from 0.1% FA-H2O (solvent A) and MeOH (solvent B), and the negative was prepared from 5 mM Ammonium acetate-H2O (solvent A:Ammonia water regulation, pH=9) and MeOH (solvent B), with a flow rate of 0.35 mL/min," there is a minor ambiguity in the sentence structure. You can clarify this by expressing it more understandably, for example: "The positive mobile phase (solvent A) was prepared using 0.1% formic acid (FA) in water and MeOH, while the negative phase (solvent B) was created using 5 mM Ammonium acetate in water (Ammonia water regulation, pH=9) and MeOH. The flow rate was set to 0.35 mL/min."

Response 14:Thank you for pointing this out. I/We agree with this comment.Therefore, we have revised the wording of the statement. This change is located in the page 4 line 151-154.

[The positive mobile phase (solvent A) was prepared using 0.1% formic acid (FA) in water and MeOH, while the negative phase (solvent B) was created using 5 mM Ammonium acetate in water (Ammonia water regulation, pH=9) and MeOH. The flow rate was set to 0.35 mL/min.] 

Comments 15:In the sentence "The mean droplet diameter, poly dispersity index (PDI), were measured," there is a plural error. Correctly: "The mean droplet diameter and polydisperse index (PDI) were measured."

Response 15:Thank you for pointing this out. I/We agree with this comment.Therefore, we have revised the wording of the statement. This change is located in the page 5 line 201.

[The mean droplet diameter and polydisperse index (PDI) were measured] 

Comments 16: The results should be analyzed in relation to the results of other researchers conducting research in the same or very similar areas. Please add this.

Response 16:Thank you for pointing this out. I/We agree with this comment.Therefore, in the results section, we added a comparison with research in similar fields. This change is located in the page 7 line 294-298; page 11 line 335-337; page 14 line 384-390.

[page 7 line 294-298: In addition to the aforementioned active substances, the extract also contains a relatively abundant variety of phenolic compounds, including quercetin, oleanolic acid and various other phenolic compounds, totaling 14 different types. These findings are in line with previous research on the polyphenolic content of lingonberry leaves[20,33].

Page 11 line 335-337: Previous studies have utilized maltodextrin and glucose microcapsules as carriers to prepare lingonberry microcapsules, achieving microencapsulation rates of 79%-81%[34].

page 14 line 384-390: Research has shown that the preparation of resveratrol as a resveratrol nanoscale emulsion resulted in a DPPH inhibition rate as high as 83.93±3.81%, an increase of nearly 13% compared to non-encapsulated resveratrol solutions[15]. Kumar etal[11] employed lecithin and Tween-80 as emulsifiers to prepare a resveratrol-loaded nanoscale emulsion, which remained stable without phase separation even after four months of storage. Under ultraviolet radiation exposure, the nanoscale emulsion significantly slowed down and inhibited the degradation of resveratrol.] 

Round 2

Reviewer 2 Report

Comments and Suggestions for Authors

This paper has been significantly improved but still has technical errors that needs to be corrected before publishing:

1) All variables throughout the text are not in italics (For example in equation 2 and equation 3). Please go through the entire text in detail and correct this.

2) In Table 2, as well as in the text, the protein concentration is expressed as a percentage. Please clarify which concentration you are talking about here.

3) Equation 2 (line 396, page 14) should be numbered as equation 4.

Introduction has been improved, so now it provides proper introduction to the topic. Materials and methods chapter as well as results have been improved. The conclusion chapter has been extremely improved.

With the corrections made, this paper has been significantly improved, but before publication all technical errors along the text must be corrected.

Author Response

3. Point-by-point response to Comments and Suggestions for Authors

Comments 1: [ All variables throughout the text are not in italics (For example in equation 2 and equation 3). Please go through the entire text in detail and correct this.]

Response 1: Thank you for pointing this out. I/We agree with this comment. Therefore, we have revised the format of all variables in the text. Modified variables in page 5 line 181 182;  page 6 line 228 229; page 6 line 237 238; page 10 line 309; Page 11 line 313 314 of the revised manuscript.

“[page 5 line 181 182: The model comprises various coefficient: β0 (intercept),βi (linear)ii (quadratic) and βij (interaction coefficients).

page 6 line 228 229: serving as the control (Acontrol). The DPPH radical scavenging activity of the sample was calculated as follows:

DPPH radical scavenging activity(%)= (1-Asample/Acontrol)×100 (2)

page 6 line 237 238: with ascorbic acid serving as the control reference (Acontrol). The ABTS radical scavenging activity of the sample was calculated as follows:

ABTS radical scavenging activity(%)= (1-Asample/Acontrol)×100 (3)

page 10 line 309:

Source

Sum of squares

df

Mean square

F-value

P-value

Model

1482.84

9

164.76

555.31

<0.0001

A

2.11

1

2.11

7.12

0.0321

B

13.13

1

13.13

44.26

0.0003

C

74.42

1

74.42

250.83

<0.0001

AB

1.7

1

1.70

5.74

0.0478

AC

1.04

1

1.04

3.51

0.1033

BC

64.16

1

64.16

216.25

<0.0001

A2

300.80

1

300.8

1013.82

<0.0001

B2

369.85

1

369.85

1246.54

<0.0001

C2

518.29

1

518.29

1746.84

<0.0001

Residual

2.08

7

0.30

Lack of Fit

0.19

3

0.062

0.13

0.9364

Pure Error

1.89

4

0.47

Cor Total

1484.92

16

Page 11 line 313 314:

Entrapping efficiency = +73.12+0.51*A-1.28*B-3.05*C+0.65*A*B-0.51*A*C+4.01*B*C-8.45*A^2-9.37*B^2-11.09*C^2

(4)

where A is the protein concentration; B is the oil phase mass fraction; C is the extract mass fraction.]”

Comments 2: [In Table 2, as well as in the text, the protein concentration is expressed as a percentage. Please clarify which concentration you are talking about here.]

Response 2: Thank you for pointing this out. I/We agree with this comment. Therefore, a detailed explanation of the mass fraction of proteins has been added to the abstract section. Modified variables in page 1 line 24 of the revised manuscript.

“[which combined soy protein isolate (SPI) and whey protein isolate (WPI) at 4% (w/w). ]”

Comments 3: [ Equation 2 (line 396, page 14) should be numbered as equation 4.]

Response 3: Thank you for pointing this out. I/We agree with this comment. Therefore, we have modified the incorrect formula changes. Modified variables in page 11 line 314 of the revised manuscript.

[Entrapping efficiency = +73.12+0.51*A-1.28*B-3.05*C+0.65*A*B-0.51*A*C+4.01*B*C-8.45*A^2-9.37*B^2-11.09*C^2 (4)]

 Additional clarifications

[I would like to express my sincere gratitude for the invaluable feedback you provided on my academic paper. Your expert insights and thoughtful suggestions have played a pivotal role in guiding my research work.

I have carefully considered each of your suggestions and made corresponding revisions to my paper in accordance with your recommendations. These modifications have not only enhanced the clarity and rigor of the paper but have also contributed to its overall quality and academic value. I believe your feedback will contribute to the further development of this research and add to its relevance in the academic and related fields.

Once again, thank you for your review and valuable suggestions. If you have any further recommendations regarding the modifications I made or areas that require further clarification or improvement, I would be more than willing to hear your input.

I extend my sincere appreciation for your invaluable contribution to my work.]

Reviewer 5 Report

Comments and Suggestions for Authors

I accept in the present form.

Author Response

I would like to express my sincere gratitude for the invaluable feedback you provided on my academic paper. Your expert insights and thoughtful suggestions have played a pivotal role in guiding my research work.

I have carefully considered each of your suggestions and made corresponding revisions to my paper in accordance with your recommendations. These modifications have not only enhanced the clarity and rigor of the paper but have also contributed to its overall quality and academic value. I believe your feedback will contribute to the further development of this research and add to its relevance in the academic and related fields.

Once again, thank you for your review and valuable suggestions. If you have any further recommendations regarding the modifications I made or areas that require further clarification or improvement, I would be more than willing to hear your input.

I extend my sincere appreciation for your invaluable contribution to my work.